# From Classical to Modern Computational Approaches to Identify Key Genetic Regulatory Components in Plant Biology

**DOI:** 10.3390/ijms24032526

**Published:** 2023-01-28

**Authors:** Juan Manuel Acién, Eva Cañizares, Héctor Candela, Miguel González-Guzmán, Vicent Arbona

**Affiliations:** 1Departament de Biologia, Bioquímica i Ciències Naturals, Universitat Jaume I, 12071 Castelló de la Plana, Spain; 2Instituto de Bioingeniería, Universidad Miguel Hernández, 03202 Elche, Spain

**Keywords:** quantitative trait loci, metabolomics, network analysis, plant breeding, proteomics, transcriptomics

## Abstract

The selection of plant genotypes with improved productivity and tolerance to environmental constraints has always been a major concern in plant breeding. Classical approaches based on the generation of variability and selection of better phenotypes from large variant collections have improved their efficacy and processivity due to the implementation of molecular biology techniques, particularly genomics, Next Generation Sequencing and other omics such as proteomics and metabolomics. In this regard, the identification of interesting variants before they develop the phenotype trait of interest with molecular markers has advanced the breeding process of new varieties. Moreover, the correlation of phenotype or biochemical traits with gene expression or protein abundance has boosted the identification of potential new regulators of the traits of interest, using a relatively low number of variants. These important breakthrough technologies, built on top of classical approaches, will be improved in the future by including the spatial variable, allowing the identification of gene(s) involved in key processes at the tissue and cell levels.

## 1. Introduction

The identification of genes to improve yield-, stress- or quality-related traits has been and still is an active field in plant science. It has traditionally relied on the generation of variability through the artificial induction of mutations (chemical or physical mutagenesis) or introgression of interesting traits from wild relatives of crops into elite cultivars, followed by intensive screening of variants throughout several seasons to obtain a stable variant. This approach did not consider the role of the mutated or introgressed gene(s) but rather focused almost exclusively on the phenotype associated to that particular mutation or introgression (e.g., traditional breeding). More recently, molecular tools contributed to facilitate the selection of potentially interesting variants at early stages, when specific DNA markers could be associated to specific phenotypic traits (canopy or root architecture, productivity, quality of fruits or other edible parts, etc.). This facilitates the breeding process but still misses the functional part. The sequencing of plant genomes has provided a blueprint on which to directly track the breeding process and understand which aspects are being modified with the introgression of genes from wild relatives, mutational events or simply by selection of the most advantageous lines. At present, several tools are available to better decipher how genes interact with each other contributing to shape the phenotype. These can be used either as marker identification or as knowledge generation tools, as they allow the identification of potentially useful genes in classical or modern plant breeding technologies (genetic transformation or CRISPR/Cas genome editing), the characterization of the hierarchy of gene expression and the reciprocal connections, hence inferring potential regulatory roles.

## 2. Classical Approaches to Identify Regulatory Components: Introduction to Marker-Assisted Breeding

### 2.1. Where It All Began: Quantitative Trait Loci (QTLs)

The two most prominent sources of variability relevant to marker-assisted breeding are (i) natural variation or (ii) random mutations induced using chemical or physical agents. These have traditionally been the main sources of variation used to identify and introgress traits of agronomical interest. The sources of natural variation are either elite cultivars or wild relatives, which often have poor or no agronomic interest per se but might carry traits of known interest (e.g., fruit quality, productivity, disease or abiotic stress resistance, etc.). However, this approach has important limitations: the genotypes’ source of the traits must be sexually compatible with the cultivars or lines of agronomic interest, and their cross must produce viable offspring on which to impose the selection process. Generally, lines are crossed by manual pollination, and the resulting heterozygous F1 progenies are subjected to multiple rounds of self-pollination using the single-seed descent method to generate recombinant inbred lines (RILs), or they are backcrossed multiple times to the parental line exhibiting the elite phenotype to obtain isogenic lines that only differ in small portions of the donor genome potentially containing the gene(s) regulating the traits of interest (Figure 1). This approach is especially interesting when the trait(s) of interest comes from a wild relative with several undesired traits, then it is necessary to reduce the genome load of the wild relative to a minimum (e.g., the introgression lines, IL, collection of *S. lycopersicum* × *S. pennellii*) [1,2,3].

Natural variation most often involves complex traits, which are regulated by an intricate network of potentially interacting genes that contribute to a specific phenotype that can be quantitated, i.e., the level or the degree of the phenotype expression. This can be correlated to the presence/absence of DNA markers spanning longer or shorter genomic regions. The presence/absence of particular DNA markers does not preclude the role of the genes present in the regions of interest. The main goal of QTL analysis is to dissect the genetic architecture of quantitative traits, allowing to simultaneously map genomic regions that significantly affect the trait and to estimate the individual contribution of those regions to the phenotypic value [4]. Markers that are tightly linked to relevant QTL can subsequently be used by breeders to guide the introgression of desirable traits into the genome of elite cultivars. In extreme cases, phenotypic differences might be primarily due to a few loci with large effects, or to many loci, each with minute effects, although the latter is the most usual situation, making marker development a daunting task. Seemingly, a substantial proportion of the phenotypic variation in many quantitative traits can be explained with few loci of large effect [5,6,7]. For example, in cultivated rice (*Oryza sativa*), studies of flowering time have identified six QTL, with the top five explaining more than 80% of the variance in this trait [8,9,10]. As was investigated later, the molecular characterization of these QTL showed that all of them encode regulatory proteins with orthologs known to be involved in flowering in the model plant *Arabidopsis thaliana* [5].

Random mutations induced using chemical or physical agents are also widely accepted as a tool to enhance crop diversity. Among the diverse chemical agents, ethyl methanesulfonate (EMS) is a chemical mutagen that induces G/C-to-A/T transition mutations in plant genomes through guanine alkylation [11]. Typically, this mutagenic compound generates point mutations that differ from one crop to another, 1 mutation per Mb in barley to 1 mutation per 175 kb in Arabidopsis or per 25 kb in hexaploid wheat [12], and it has been widely used in forward genetics as a source of random variability arising from a highly homogeneous population (e.g., seeds from a single Arabidopsis plant, a cell culture obtained from a single genotype and tissue, etc.) [13].

Among the physical agents, classical radiation-induced mutagenesis using high-energy particle radiation such as X-ray and gamma ray are widely used because they could induce a large number of genomic mutations [14]. However, emerging mutagens, such as accelerated heavy-ions or protons, have the advantage of inducing high mutation frequency (and a broad mutation type spectrum) at relatively lower doses than classical irradiation treatments, causing a large amount of damage to DNA in a small area [15]. An important advantage of physical over chemical mutagenesis is that it induces mutations that are substantially more likely to damage gene functions (e.g., the partial or complete deletion of a gene), thus producing more gene loss-of-function mutations related to target traits with fewer mutations per genome [14]. Following mutagenesis, the mutagenized population must be thoroughly screened to identify interesting mutants in terms of phenotypic response (stress resistance, plant architecture, etc.). It is important to notice that each mutation is the result of a rare, random event that is unlikely to occur multiple times in the mutagenized population. Therefore, mutants with similar phenotypes typically arise from different mutation events.

**Figure 1 ijms-24-02526-f001:**
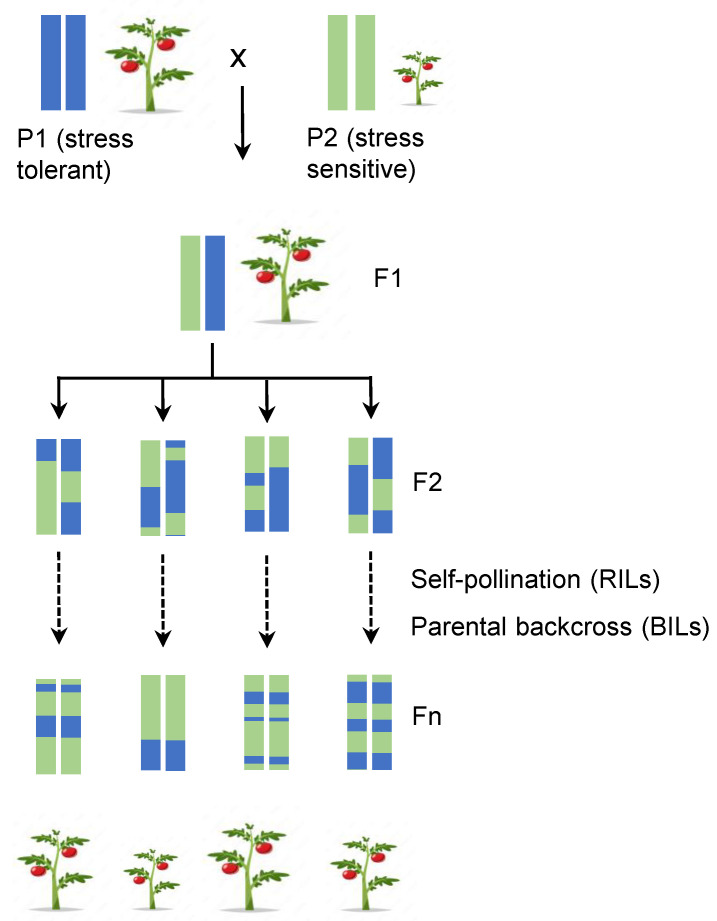
Schematic representation of breeding scheme to identify QTLs conferring stress tolerance using recombinant inbred lines (RILs) by self-pollination or backcross inbred lines (BILs) by crossing several times with one of the parental lines (P1 or P2) and subsequent self-pollination during several generations (Fn) until the population was brought close to homozygosity [16].

In the early days, breeders mostly employed random DNA markers derived from genomic regions that are closely linked to the gene of interest. The main drawback of these markers is that their predictive value depends on the known linkage between marker and target locus [17]. The popularization and widespread commercialization of Sanger-based automated capillary sequencers about two decades ago [18] led to a boost in the development of various types of DNA-based markers, such as microsatellites, random amplified polymorphic DNA (RAPD) markers, amplified fragment length polymorphisms (AFLPs) or single-nucleotide polymorphisms (SNPs). The combination of these markers and bulked segregant analysis (BSA) allowed the development of “chromosome landing” protocols for the rapid identification of markers tightly linked to a trait of interest [19], helping breeders to bridge the gap between genotype and phenotype. Next-generation sequencing (NGS) technologies enabled the development of genotyping-by-sequencing methods [20] and the rapid sequencing of plant genomes, thus contributing to enhance the resolution of QTL mapping and the popularization of genome-wide association studies (GWAS), in which the association between a trait of interest and thousands of SNP markers is tested. More recently, the advent of third-generation sequencing technologies has enabled near-complete, chromosome-scale assemblies of whole genomes using long reads produced by Pacific Biosciences (PacBio) and Oxford Nanopore Technologies (ONT) sequencers, either alone or combined with short reads from Illumina, which are sometimes used to correct sequencing errors present in the long reads [21,22]. The high-fidelity (HiFi) long reads produced by PacBio’s circular consensus sequencing (CCS) method are now routinely used for the assembly of highly contiguous plant genomes [23]. These genomic sequences with unprecedented quality offer an ideal reference for genome resequencing experiments, where the short reads produced by the Illumina technology are aligned to a reference genome using programs such as BWA and Bowtie2, which use algorithms based on the Burrows–Wheeler transform [24]. The resulting alignments can then be scanned by variant/SNP calling tools, such as BCFtools, FreeBayes, VarScan2 or the Genome Analysis Toolkit (GATK) package [25], to identify polymorphisms that can subsequently be used in mapping-by-sequencing, QTL-seq, or GWAS experiments. However, the identification of markers closely associated with trait-governing genes still remains a challenging task, as high-coverage genotyping for crops with large genomes and the requirement of measuring phenotypes for large numbers of individuals are economically costly. This limitation is linked to the requirement of QTL analysis of large population sizes, as hundreds of individuals must be accurately genotyped and phenotyped under relevant environmental conditions. To overcome this limitation, QTL-seq, which combines NGS and BSA, is becoming increasingly popular [26]. In this method, only two pools of plants exhibiting opposite, extreme phenotypes are sequenced, and QTLs are then identified by finding SNPs whose allele frequencies differ significantly between the two pools (Figure 2). To enable the detection of QTL using strategies based on BSA, different software tools and at least nine different statistics have been developed, which have been reviewed in [27]. However, like in classical QTL mapping, these methods will only allow detecting QTLs for which genetic variation between the parental lines exist. Hence, because parental lines are unlikely to contain segregating alleles of every locus contributing to the trait, some important genes will remain undetected.

Interestingly, QTL mapping can also be applied to the study of gene expression levels, allowing researchers to gain insight into the genetic architecture of the variation in gene expression and identify regulatory genes that control the expression of the trait of interest in plants. Initial studies involving so-called “expression QTLs” (eQTLs) were based on the results of microarray hybridization experiments [28,29,30]. More recently, the modern massively parallel RNA sequencing (RNA-seq) techniques have opened the door to correlate the expression level of each individual gene expressed in a given tissue with the genotype of thousands of molecular markers, particularly SNPs, which can be detected in the same experiment [31]. In order to map and detect eQTLs, expression levels are analyzed with methods similar to those applied to map QTL, underlying other quantitative traits, such as size or yield. In these studies, the correlation between gene expression levels and other quantitative phenotypes might also be detected. Allelic variation at the identified eQTLs affects the expression of other genes, and hence, it might facilitate the identification of genes that control phenotypes of interest. The genes whose expression is affected by eQTLs, called e-genes, are also identified by eQTL mapping studies. Some eQTLs (*cis*-eQTLs) affect the expression levels of genes located in their vicinity (i.e., genetically linked to the molecular marker or polymorphism that allowed their detection), while other eQTLs (trans-eQTLs) affect the expression of unlinked genes [32]. So-called transcriptome-wide association studies (TWAS) have helped to identify correlations between a quantitative phenotypic trait and polymorphisms that co-localize with a *cis*-eQTL [33]. While *cis*-eQTLs might correspond to sequences containing regulatory elements, such as promoters, enhancers, or transcription factor binding sites, trans-eQTL can identify regulatory proteins, which include not only transcription factors but also other trans-acting regulators that potentially affect the expression of many other unlinked genes in the genome. These trans-eQTLs often correspond to master regulators of developmental or metabolic pathways [30]. A genomic region containing a cluster of trans-eQTLs, which affect the expression of a large number of genes, is referred to as an eQTL hotspot. Because eQTLs can potentially identify both regulatory genes and their targets, these experiments can help to build regulatory networks for the traits of interest, which can be integrated with the information obtained by studying the co-expression of genes.

### 2.2. Marked Assisted Selection

The above-described experimental approaches allow the identification and mapping of QTLs that contribute to a trait of interest. This enables the use of tightly linked molecular markers to guide the introgression of QTLs into an elite parent line to develop near-isogenic lines (NILs). Introgression can result in the so-called “mendelization” of QTLs, which are transmitted in a predictable manner as single Mendelian factors, facilitating their fine mapping and cloning, as well as their use in marker-assisted breeding programs [34]. In crops, the mendelization of loci controlling quantitative traits facilitates tracking their presence with molecular markers, which is the basis of marker-assisted breeding. Moreover, using molecular markers (in any of their forms) will benefit from the parallel use of classical phenotypic selection, possibly through the use of weighted selection indexes, because no molecular marker will explain 100% of the total variance of a target trait, and there might be pleiotropic effects that are difficult to take into account. By using molecular markers to track the transmission of specific QTLs in a segregating population, breeders can more easily identify and select for individuals with the desired traits. This information can be used to optimize the timing and design of crosses between different genotypes. QTLs of major effect often correspond to regulatory genes controlling metabolic pathways or developmental processes with a noticeable effect on crop yield or other aspects of plant biology, such as resistance to pathogens or tolerance to various types of stress, which are highly desirable traits in crop plants. Similarly, the study of eQTLs in model and crop plants has furthered our understanding of the molecular basis of traits of agronomic interest. In *Arabidopsis thaliana*, a QTL spanning the MYB28 transcription factor gene affects both the content in aliphatic glucosinolates and the expression levels of genes involved in their biosynthesis [35], demonstrating how QTLs can have a significant impact on the production of defense compounds against herbivores. In two studies involving cotton, the overlapping locations of eQTLs and loci identified in GWAS experiments helped identify candidate genes controlling fiber quality, growth and salt tolerance [36,37]. In these studies, the integration of results from GWAS experiments and eQTL mapping has helped researchers to focus on the loci that were more likely to control the trait of interest.

## 3. Annotation of Genes: Role of NGS and Comparative Genomics

As mentioned above, during the last decades, a massive revolution has happened at the level of DNA sequencing with the development of NGS technologies. For instance, the sequencing cost of the human genome until 2007 was around USD 10 million but has experienced a 4000-fold drop since the advent of NGS sequencing platforms, and now it is possible to sequence the 3200 Mb human genome with 30× sequencing depth for about USD 1000 (https://www.genome.gov/about-genomics/fact-sheets/DNA-Sequencing-Costs-Data, accessed on 23 December 2022). Plant genome sizes vary several orders of magnitude from the 60 Mb size of the carnivorous corkscrew plant *Genlisea aurea* genome to the astonishing 152,000 Mb size of the Japanese plant *Paris japonica* genome [38]. However, plant genome sequencing can be even more complex because of the polyploidization process, which is a frequent event throughout plant evolutionary history, and has been associated with plant domestication [39]. Until now, 1031 genomes of 788 different plant species (including subspecies and cultivars) have been sequenced and published [23], still more than 2000 genome sequences are available at the Genome database of the National Center for Biotechnology Information (NCBI) (https://www.ncbi.nlm.nih.gov/genome, accessed on 23 December 2022). In addition, the reduced cost and high coverage of high-throughput RNA-seq has allowed it to be the most popular technology for profiling plant gene expression. As a result, the number of plant RNA-seq datasets has been increasing exponentially to the ~83,000 datasets collected at the NCBI Bioproject database (https://www.ncbi.nlm.nih.gov/bioproject/, accessed on 23 December 2022), and for some major crops, such as maize, rice, soybean, wheat and cotton, the plant community has collected a total of ~45,000 libraries so far [40]. Another aspect to consider is the tremendous increase in the quality of the sequenced genomes and the concomitant improvement in gene annotation associated to the advances in sequencing technologies. In fact, assemblies made using modern long-read technologies such as PacBio or ONT show a ~32-fold increase in the mean contig N50 (the length of the shortest contig in the set of contigs containing at least 50% of the assembly length) compared with short-read technologies [41]. This is especially crucial when dealing with complex genomes such as highly heterozygous diploid genomes or polyploid genomes, allowing the development of specific pipelines such as Genomescope 2.0 and Smudgeplot [42].

Hence, a vast amount of plant genetic information has been assembled using well-established bioinformatics pipelines based on the overlap–layout–consensus (OLC) and De Bruijn graph (DBG) paradigms, which have allowed for gene functional annotation [23,43,44]. In general, for gene annotation, most researchers combine ab initio gene detection with the alignment of genomic and transcript data and known gene sequences from related species. Some of the most popular tools to structurally annotate a genome include the automated pipelines MAKER2, MAKER-P (which was specifically developed for plants) BRAKER1, Trinotate, GeneMark-ET or AUGUSTUS, among others. For instance, the latest annotation of the Arabidopsis genome, Araport 11, allowed the identification of 27,655 protein-coding and 5178 non-protein-coding genes [45]. Next, the annotated genes must be functionally classified by inferring their function according to their sequence similarity using databases based on experimentally derived knowledge, such as the Gene Ontology (GO) database (http://geneontology.org/, accessed on 20 December 2022), which provides a collection of terms to precisely describe the function of each gene. Actually, the functional GO annotation of protein-coding genes is the most widely used, and it ranges from the functional annotation of more than 94% of the Arabidopsis genes to around 60% of genes annotated in other less well-studied species such as potato or sugarbeet [45,46,47]. Hence, for most plant genomes, the use of other resources is a must to obtain a more complete functional annotation, such as the Kyoto Encyclopedia of Genes and Genomes (KEGG; http://www.kegg.jp/, accessed on 20 December 2022) that aims to link genomic- and molecular-level information to higher-level functions of the cell, organism and ecosystem, or the Plant Metabolic Network (http://www.plantcyc.org, accessed on 20 December 2022) that is mainly used to describe enzymatic functions and build up reaction networks. Really interesting for the identification of functional gene families are the different resources developed for functional identification based on protein domain similarity within a sequence. Some of the more popular ones have user-friendly web interfaces, such as the InterPro database (https://www.ebi.ac.uk/interpro/, accessed on 18 December 2022) or the Conserved Domain Database (https://www.ncbi.nlm.nih.gov/cdd/, accessed on 18 December 2022).

The screening of transcription factors as master regulators of several, often interconnected, plant processes is particularly interesting. This is possible because typical or conventional transcription factors (TFs) interact with DNA in a sequence-specific manner through one or more well-defined DNA-binding domains (DBDs). So far, more than 80 different DBD types have already been identified in eukaryotes. TFs are usually classified into superclasses and families according to the structural relatedness of their DBDs, which normally provides clues for their TF function [48]. Hence, the putative TFs can be identified based on the presence of conserved DBDs or on the sequence similarity to previously characterized transcription factors. Vice versa, sequence-specific DNA binding is the main and first feature that is commonly addressed while trying to characterize (or discover) a new TF. The high quality of genomes assembled using the long-read sequencing technologies has allowed the accurate determination of *cis* elements, which increase our knowledge of TFs’ functionality and the different plant responses they control [49]. Therefore, the characterization of *cis*-binding DNA motifs in the promoter sequences of differentially expressed genes (DEGs) might also contribute to identify stimulus-dependent gene expression (HOMER Motif analysis software, http://homer.ucsd.edu/homer/motif/; PlantPAN 3.0, http://plantpan.itps.ncku.edu.tw/, both accessed on 20 December 2022).

Moreover, publicly available bioinformatic resources such as InterPro, Pfam and SUPERFAMILY provide curated DBD models describing the amino acid sequences of groups of conserved polypeptide regions and domains that could be scrutinized. For instance, OMICSBOX software (https://www.biobam.com/omicsbox/, accessed on 20 December 2022), formerly called Blast2go, searches for conserved domains or sequence similarity of translated proteomes from annotated genomes or transcriptomes by a BLAST-based approach into reference bioinformatic resources, producing the functional classification of these genes. However, some DBDs and their sequence models may be promiscuous and produce false-positive hits to non-TF proteins when blasted, and there are also some TFs that display sequence-specific DNA-binding activity without any recognizable or standard DBD, making the correct functional annotation not so straightforward, requiring experimental functional validation [48].

## 4. Correlation of Genes and Traits Using Omics Technologies

The advent of omics technologies including sequencing-based transcript profiling, shotgun proteomics, metabolomics and automated phenotyping of traits such as plant architecture, height and leaf area, as well as physiological parameters (photosynthesis, water content, etc.), has provided scientists with abundant data and also brought in the issue of complex dataset interpretation [50]. One of the most popular analyses to handle large datasets of omics data is co-expression analysis, which enables the identification of pairs of variables (mRNA, miRNA, metabolites, etc.) with a correlated expression across several samples (genotypes, treatments, time points, etc.); see Figure 3. The more conditions (ideally orthogonal), the more powerful the method is. The degree of correlation is expressed as a score value which reflects the degree of similarity of the “expression” pattern between two variables. A score value above a certain threshold is defined as a sign of co-expression. All variable pairs showing score values falling within certain limits (either positive or negative) can then be used to construct a network in which clustering of variables is interpreted as a result of a coordinated regulation [50]. This analysis is a powerful tool to investigate interactions between different biological processes, identification of potentially key regulatory elements and also to predict functions of unknown genes for which a functional characterization is not available yet [51].

### 4.1. Glossary of Network Analysis

**Co-expression network:** This refers to a set of (more or less) densely interconnected variables in which the degree of connectivity is linked to similarity in expression profiles, abundance or intensity of a given variable throughout the samples (genotypes, conditions, time series, etc.). These usually express gene expression data and metabolite or protein accumulation.**Edges and nodes:** In a network, the variables are nodes or vertices and are usually depicted as points. The connections between the nodes are referred to as edges and are usually depicted as lines between points.**Module:** A cluster of highly interconnected (showing high absolute correlation, either positive or negative) variables (genes, metabolites, proteins, etc.) that potentially reflects functional similarities among cluster members. Modules can be further refined by applying GO or pathway enrichment criteria.**Connectivity:** The correlation existing between pairs of variables, inferred from correlation- or mutual-information-based methods.**Module eigengene E:** Defined as the first principal component of a given module, it is a representation of the variable expression profiles in a module. These values can be correlated to an external trait (e.g., phenotype). It is also related to the **module membership** by correlation of the variable expression level with the module eigengene E; values close to 1 or -1 indicate positive membership.**Hub:** It is generally defined as a “highly connected gene or protein” which is a member inside co-expression modules. The topology of a hub might reflect its role as a regulatory element.**Module significance:** Absolute average variable significance within a module, which is determined by correlating variable expression to an external trait (e.g., phenotype).

Networks constitute powerful mathematical representations of the interactions among different biological components to model biological systems which are extremely complex in nature. From this point of view, a network analysis can be fed almost with anything, and it will surely find correlations between pairs of elements that can be subsequently represented as a network of interactions. The network is essentially constituted by variables represented as nodes and the interactions among those nodes, derived from an iterative pairwise correlation analysis, represented as edges. Edges can represent positive or negative interactions, and nodes can occur grouped together, as a cluster or a module, which suggests a common functional role, or separated. Separate nodes with a high degree of interaction are known as hubs and might identify key regulatory elements controlling the information flow from the stimulus to the response. The network itself is a graph that can be analyzed using different algorithms to gain insight into the network architecture, defining functional modules and hubs that connect different modules [53]. The network architecture, the interaction between nodes (variables) and their distance, which is related to the degree of interaction, as well as the direction of the interaction, either positive or negative, can be subsequently corrected by using available data from empirically assessed interactions at data repositories (Table 1).

Genes belonging to the same metabolic pathway are expected to be subjected to temporal and spatial co-regulation at the level of mRNA abundance, thus reflecting the functional coordination and collaboration to produce metabolites [63,64]. Likewise, orthologous genes from different plant species are also expected to behave similarly under comparable experimental conditions, in line with their evolutionarily conserved gene functions. There are several examples in the literature of conserved co-expression modules in homologous or orthologous genes related to different plant processes, such as photosynthesis, seed longevity or cell wall biosynthesis across plant species [53,65]. To date, more than 300,000 sequenced RNA samples are available from public repositories, corresponding to several thousands of experiments encompassing gene expression in different organs, tissues, developmental stages and experimental treatments for several plant species (Table 1) [66]. This unprecedented amount of information along with the integration of gene expression in an anatomic, experimental and temporal context in easy-to-grasp visualization tools facilitates understanding of not only gene function but also of how gene expression is orchestrated, pointing to potential key master regulators.

As a major drawback, the generation of co-expression data requires that *omics* datasets are properly normalized. To this respect, transcript profiling data can be found in several formats:log_2_-fold change between treated samples and controls, which facilitates identification of over- and down-regulated genes.Reads per kilobase of transcript per million mapped reads (RPKM) for single-end reads from RNA-seq experiments [67], which facilitates comparison of transcript levels within and between samples.Fragments per kilobase of transcript per million mapped fragments (FPKM) for RNA-seq experiments producing paired-end reads.Transcripts Per Million (TPM), which is similar to the former two, but the order of operations is inverted.Trimmed Mean of M-values (TMM) for genes meeting a corrected *p*-value and false discovery rate (FDR) lower than 0.05, which dramatically reduces the number of false discoveries due to different distribution of expressed transcripts [68]. This method assumes that the most genes are not differentially expressed.

Other omics that can be integrated into co-expression network analysis are metabolomics. Metabolomics data can also be presented in several ways: absolute values, which is less common, and relative values (peak area relative to internal standard area, total intensity, sample amount, etc.), which are more widespread but might differ between techniques, instruments, extraction procedures, etc. Moreover, despite values within the same batch of analyses or from the same laboratory being highly reproducible and robust, they differ greatly when considering a different platform (operators, instruments, solvents, etc.). In this regard, the standardization of procedures for metabolomics is less advanced than for RNA-seq, despite great efforts having been made to provide a set of rules to report metabolomics data, such as the Metabolomics Standardization Initiative or MSI [69]. The preferred metabolomics platforms are based on mass spectrometry (MS) measurements coupled to chromatographic or capillary electrophoresis separation, followed by nuclear magnetic resonance (NMR), usually not coupled to any separative technique. These analytical techniques generate data with different appearance and scaling; therefore, they need to be normalized before attempting any statistical analysis. The normalization method is not trivial, as it must be noted that metabolite concentrations correlate better with metabolic fluxes than with enzyme expression levels. This fact has been attributed to reaction mechanisms, the self-regulatory nature of metabolic networks, post-translational regulation and the topological organization of metabolism [70]. As early as 2005, Hirai and co-workers [71] published a study in which the integration of metabolomics and transcriptomics allowed the identification of regulatory genes of different metabolic pathways such as anthocyanin and glucosinolates. Authors used Batch-Learning Self-Organizing Maps (BL-SOM) to attain this objective. The normalization of transcript and metabolite profiling data (microarray and different targeted and nontargeted analytical techniques) was attained by calculating the logarithm of the ratio of treated vs. control samples. Hence, both metabolomics and transcriptomics data exhibited similar values, removing any effect of variations in sample amount. Nowadays, most normalization methods are data-based; to this regard, it must be taken into consideration that both sample and variable normalizations can either reduce or increase analytical variance and batch effects [70]. Normalization strategies have two main objectives: preprocess metabolomics data for a subsequent statistical analysis and the removal of batch effects. Batch effects are especially relevant in MS-based metabolomics; therefore, different strategies have been tested and implemented: LOESS (Locally Estimated Scatterplot Smoothing) using quality controls (QC) interspaced in the batch sample list, the variation of QC values across sample batches is taken as a proxy to evaluate instrumental drift; and post-acquisition data normalization using MS useful signals or probabilistic quotient normalization (PQN), which prevents impact of variability in concentration. For ^1^H-NMR-based metabolomics, PQN [72] is based on the calculation of the most likely dilution factor by looking at the distribution of the quotients of the amplitudes of a test spectrum compared to that of a reference spectrum whereas constant sum (CS) simply normalizes total spectrum intensity to a single value [70]. Other strategies for cross-sample, between-sample (e.g., sum, median, weight and quantile) and within-sample normalization (e.g., feature transformation) are also widely used in the field of nontargeted metabolomics. Despite the active research in this field, to date, there is no definitive and standardized methodology. The sequential application of a normalization strategy should be dependent on the metabolomics platform and ensure that it does not get rid of the biological information or the variance associated to the sample [70]. More recently, Correia and co-workers [73] investigated several workflows, with their respective normalization strategies comprising large homogeneous and small heterogeneous datasets, and concluded that the biggest impact on network construction was related to between-sample normalization.

Proteomics can also be integrated within a co-expression network analysis and, as in metabolomics, different platforms are also available and widely used: 2D-gel electrophoresis to investigate differentially expressed proteins (e.g., DIGE, [74]), where the identification of proteins is attained by performing offline mass spectrometry analyses to match with spectra available in databases (Uniprot, swissprot, pfam, etc.), and shotgun proteomics based on analysis of samples by liquid chromatography coupled to mass spectrometry, which allows multiplexing through isobaric labeling of peptide extracts and co-injection [75]. Both approaches allow the analysis of differentially expressed proteins and also the identification of protein post-translational modifications through changes in *mz* associated to the incorporation of different biologically relevant moieties (e.g., acetylation, phosphorylation, glycosylation, sumoylation, etc.). MS-based proteomics approaches can be normalized as for metabolomics. For instance, Minadakis and co-workers [76] scaled protein abundance data, derived from the sum of ion count for each of the peptides associated to an individual protein, between −1 and +1 to integrate proteomics and transcriptomics in a co-expression network to investigate protein and gene changes in response to diurnal rhythms in cyanobacteria. Another example is the ProtExA tool [77], which primarily uses log_2_-transformed datasets (as a requirement of the LIMMA package, used for the statistical analysis, that is, implemented into the workflow) but can also use several other normalization methods. Therefore, it is likely that the statistical approach chosen might coerce the normalization strategy, which is something to take into consideration when interpreting results. Cueff and co-workers [78] used straightforward centered and scaled 2D-gel proteomics data to build a co-expression network to investigate secondary dormancy induction by hypoxia or high temperature in barley seeds, but no integration of other omics was performed.

### 4.2. Co-Expression Network Analysis

As mentioned above, networks constitute a powerful mathematical representation of the interaction among different biological components to model naturally complex biological systems. The network is essentially constituted by variables represented as nodes and edges that represent the interactions among those nodes, either as positive or negative, or close (high correlation) or loose (low correlation) interactions. Individual nodes exhibiting a high degree of interaction are known as hubs and are considered potential key regulatory elements controlling the information flow from the stimulus to the response. It is important to note that, despite being essentially a nonsupervised approach, accurate annotation of nodes, as well as grouping according to function of pathway, is a necessary step to contextualize resulting networks and to remove spurious correlations with no clear biological relevance. In this sense, co-expression network analysis is an excellent strategy to uncover novel and unanticipated interactions between well-characterized functional modules, to elaborate data-driven hypotheses on the key regulatory role of different elements and to hypothesize gene and other molecular functions of uncharacterized nodes based on their surrounding functional landscape. In addition, different software programs exist for integration of omics and network construction and analysis [79]. Some examples are listed in Table 2.

### 4.3. Construction of a Network

The network can be understood as a 3D unrooted dendrogram in which distances between nodes are calculated as a function of variable expression correlation. Therefore, the first step in the network construction is the generation of a dataset containing the distance metrics between pairs of variables representing steady-state or time series kinetics. There are several methods for gene network inference, including correlation, mutual information (MI), Bayesian network and probabilistic graphical models. Typically, correlation and MI methods are used for constructing large-scale graph convolutional networks (GCNs) with more than 10,000 nodes [88]. The most popular and straightforward methodology to generate a list of pairwise comparisons between variables is Pearson’s correlation. With this methodology, a correlation coefficient and a *p*-value are generated for each pair of nodes. It has the advantage of being robust, fast (several millions of combinations in a dense dataset can be calculated within seconds in any benchtop or laptop computer) and easy to implement [89] but, unfortunately, it can only detect linear relationships [90]. Moreover, Pearson’s correlation is sensitive to outliers, leading to the false discovery of correlations when extreme values appear in the variable dataset. Conversely, Spearman’s correlation metrics enables the identification of nonlinear correlations [91] through the representation of correlation ranking between pairs of variables [50], and it is more robust to extreme values that “force” high correlation indices. The qualitative biclustering algorithm (QUBIC) is another method that enables capturing co-expressed modules under a subset of all the conditions without prior information to group the datasets. As a drawback, it requires large numbers of sample sets representing the different conditions to be efficient [92]. Other methods, known as MI, generate a generalization of pairwise correlation coefficients, which detects statistical dependence between two variables. These methods can be further improved, enabling the identification of both linear and nonlinear relationships [89]. However, the selection of the statistical method should be defined by the biological question to answer. In this regard, several attempts to empirically evaluate and select the best distance metrics and inference methods have been carried out. For instance, Huang and co-workers [88] tested several distance metrics: Pearson, biweight midcorrelation, Spearman, Kendall rank correlation coefficient, Gini correlation coefficient and cosine similarity coefficient, as well as MI-based methods: ARACNE (additive and multiplicative), MRNET and CLR on microarray and RNA-seq data from maize. In this work, correlation-based metrics resulted in a more predictive co-expression network, although interactions with some specific genes were better detected with MI-based methods [88]. Similar results were obtained by Lieske and co-workers [93] with *Arabidopsis thaliana* microarray and RNA-seq datasets, particularly focusing on well-known metabolic pathways. These authors reported that Pearson’s correlation combined with Highest Reciprocal Ranking (HRR) performed better than other correlation metrics or MI-based methods. One of the most popular and widespread methods used to perform network analysis is weighted gene co-expression network analysis (WGCNA), which can be applied with the WGCNA R package [94], which performs network construction, module selection, module and gene selection, calculations of topological properties, data simulation and visualization and interfacing with external software packages. Within this package, different co-expression measures (such as Spearman or biweight midcorrelation) are implemented apart from Pearson’s correlation.

### 4.4. Module Selection

Once the network is already constructed, the next logical step is to proceed with module selection containing elements potentially sharing functional similarity (same signaling or metabolic pathway, etc.). To define a module, several measures of network interconnectedness have been defined [95]. As a default method, the WGCNA R package uses the topological overlap measure, or TOM. Essentially, modules can be detected by performing unsupervised clustering using hierarchical cluster analysis, or HCA. Then, branches of the dendrogram correspond to modules that can be identified using different methods: constant-height cut or dynamic branch cut methods [94]. The number of clusters depends on the selection cutoff value, which is defined after a cluster stability/robustness analysis. It must be noted that large datasets are more likely to generate artifactual connections or edges. Therefore, to improve the biological meaning of networks, threshold values need to be calculated to ensure network properties and reduce false associations. To this respect, Burns and co-workers [96] suggested a stepwise approach that has already been implemented in the software Knowledge Independent Network Construction (KINC, https://kinc.readthedocs.io/en/latest/, accessed on 21 December 2022). However, it must be noticed that these stringent values (around 0.85–0.95) exclude moderate relationships, which usually underlie extremely complex biological questions. This includes missing values which might have a biological meaning and that should be considered in the association tests.

As already mentioned, GO term enrichment or other biological information tests (such as metabolite over-representation analysis extracted from KEGG, Reactome, BioCyc or AraCyc, etc.) are a highly recommended step to extract biologically meaningful information from networks. To facilitate visualization and summarizing, the WGCNA R package implements a function to extract eigengenes of each module, which can be interpreted as the weighted average expression profile of a given module [94]. Node module membership usually follows a binary assignment in HCA, as well as most standard clustering methods, and it is usually sufficient for most studies, but, for some applications, a fuzzy measurement of module membership for all nodes might be advantageous when nodes that lie near the boundary of a module or are intermediate between two or more modules are expected.

In addition, performing network construction and module detection in large datasets, especially when spanning different omics datasets, might be computationally challenging when operating with small benchtop or laptop computers, even for “light” operations such as Pearson’s correlation. For this reason, the WGCNA R package has implemented a function that preclusters nodes into large clusters, known as blocks, using a variation of k-means clustering, and subsequently applies HCA to each block. Modules are then defined as branches of the resulting dendrogram. Then, to integrate the module detection results across blocks, an automatic module merging step between modules with highly correlated eigengenes is performed. An interesting option when dealing with different networks and their respective adjacency matrices is the identification of consensus modules, present in a big fraction of all networks, further supporting connectivity between nodes and the identification of hubs [94].

Biological significance can be encoded numerically, the greater the significance the greater the number. In other types of analyses, this significance can be interpreted as pathway membership or functional relationship. This can be achieved by using a sample trait to define omics based on the absolute correlation between the trait and the omics profile data. Moreover, module significance can also be defined as the average gene significance across module genes, using eigengene E(q) and correlation or *p*-value resulting from a univariate regression between E and the sample trait, generally a continuous trait. As a result, modules with high trait significance (correlation coefficient and/or *p*-value) may represent modules related to the sample trait and, hence, genes with high module membership are good candidates for further experimental validation of the gene–trait association. Network topological properties are interesting aspects to analyze and describe, which constitute the network statistics or indices, namely: whole network connectivity, intramodular connectivity, topological overlap, clustering coefficient, density, etc. Indeed, differential analysis of network concepts such as intramodular connectivity is linked to specific regulatory changes affecting the expression of different omics data profiles. The WGCNA R package has several functions already implemented to attain topological network analysis. Finally, one of the most attractive outputs of the network analysis is probably the network visualization, as well as the possibility to manually manipulate it to extract information.

One of the most powerful software to attain network visualization is Cytoscape, an open-source platform which includes a plethora of community-contributed plugins (now called apps) to carry out different relevant analyses in molecular life sciences (e.g., BINGO, stringAPP, CluePedia, CoExpNetViz and many others). It can be freely downloaded from https://cytoscape.org/ (accessed on 21 December 2022) and is based on JAVA^TM^, which enables its usage in different computer platforms. Cytoscape was developed back in 2001 to provide biologists from different areas with an interactive tool that allows a close manipulation and inspection of the constructed network (zooming in and out, moving or removing nodes, etc.). Moreover, Cytoscape nowadays encompasses several apps that expand the capabilities of the software, with tools such as functional annotation and discovery, module detection and analysis of different topological attributes of the network, even performing differential network analysis to investigate potential rewiring of network connections in response to different factors [97], in addition to all the visual customization tools available. Therefore, it is usually more convenient to use Cytoscape to visualize and analyze topological attributes of networks constructed using other tools (e.g., WGCNA R package). Indeed, it is possible to seamlessly connect R with different external tools including Cytoscape, such as the RCX tool [98] or, conversely, use Cytoscape from within R, such as RCy3 [99].

Another interesting software package is mixOmics (http://mixomics.org/, accessed on 20 December 2022) (currently only available from bioconductor) [80], which performs multivariate analysis of biological datasets focusing on data exploration, dimension reduction and, particularly, visualization. This software, implemented in R, is especially aimed towards the integration of different biological data sources, which the method assumes have been appropriately normalized to transform discrete into continuous data modes (microarray, RNA-seq, MS-based proteomics or metabolomics, 16S rRNA sequencing for meta barcoding, etc.). In the mixOmics workflow, a data matrix with N observations (typically distributed in rows) × P predictors (this is normalized omics data) and a categorical outcome (e.g., control and treated, genotype 1 and genotype 2, etc.) is expressed as an indicator matrix, where columns represent each category (genotype, treatment, etc.) and rows indicate class membership or categorization value. As indicated by authors, the software can handle several thousands of predictors, but to optimize computational time, it is highly recommendable to thin predictors to less than 10,000 by, for instance, removing low-count genes in RNA-seq data or predictors with zero variance across observations to optimize computational time. To reduce data dimensionality, the software has implemented a series of multivariate analysis strategies: unsupervised analyses, such as principal component analysis (PCA), independent component analysis (ICA), partial least squares regression (PLS), multigroup PLS, regularized canonical correlation analysis (rCCA) and regularized generalized canonical correlation analysis (rGCCA), and supervised analyses, such as PLS-DA, GCC-DA and multigroup PLS-DA [81]. In addition, mixOmics provides sparse variants which allow feature selection and, hence, the identification of key predictors related to the molecular signature. Using this approach, Hasbún and co-workers [100] recently identified secondary metabolism rearrangement as a key response that allows primed *Pinus radiata* seedlings to thrive under stressful conditions. This was achieved by integrating proteomics data as predictors and the physiological measurements as response using sPLS multivariate models. The integration of different omics data measured on the same biological samples (N-integration) is also possible. This is performed with the DIABLO (Data Integration Analysis for Biomarker discovery using Latent cOmponents) method, which identifies a multiomics signature that discriminates the outcome of interest. Essentially, DIABLO identifies a signature constituted by highly correlated features across different omics by modeling relationships between the omics datasets. To attain this, linear combinations of variables are constructed that maximize the sum of covariances between pairs of datasets. The design matrix indicates the weight of each pairwise covariance. The indicator or response variable is transformed into a dummy variable within the function. Finally, a regression is performed through sparse GCCA to compress each dataset. The implementation and application of the DIABLO method has proved useful to identify already known and novel multiomics biomarkers (e.g., mRNAs, miRNAs, CpG islands, proteins and metabolites) [101]. Each dataset (each omics) is represented as a block, which are then inter-correlated and represented as a heatmap to show the profile of each descriptor, as well as a Circos plot that depicts the correlation between predictors (estimated from latent components as explained in [102]). Finally, this is represented as a network to identify modules of closely related features. On the other side, it is also possible to integrate the same descriptors across several independent studies or P-integration using MINT (Multivariate INTegrative method) [103]. By following this approach, sample size is increased, hence allowing comparison among similar studies, thus providing a benchmark for a specific condition, cultivar, etc., although the primary objective is the classification of samples, subsequently providing a robust molecular fingerprint associated to specific sample groups. Essentially, the methodology is similar to that used in DIABLO; the number of components that describe the biological system is defined by sparse PLS-DA, which identifies the molecular signature that can be the used to build the model, taking into consideration the balanced error rate (BER) calculated as the averaged proportion of wrongly classified samples in each class, weighting up small sample size classes. Cross-validation as “leave one out” is performed by removing a particular study only once, reflecting the reality of prediction performed on independent external studies and based on a reproducible molecular signature identified on the training set [103]. A more detailed list of other available tools are referenced in [79], and the most interesting are listed in Table 2.

## 5. Applications in Plant Biology

The most straightforward application of co-expression network analysis is the study of metabolic pathways and their regulation [104,105,106,107], including plant stress responses [108,109,110] and, more importantly, the identification of potential candidates for the biotechnological improvement of crops [51]. This is of special relevance, as it makes it possible to directly transfer all knowledge gained in model species over the years to species of agronomic interest, such as soybean [108]. In this species, a co-expression network was constructed from time series RNA-seq data using a correlation-based method, which subsequently was imported into Cytoscape for further analyses, such as module and hub gene identification, to characterize the salt stress response in a sensitive and a tolerant cultivar. A similar approach has also been used to investigate the mechanistics of a physiological disorder in citrus, known as juice sac granulation, associated with huge crop losses in pummelo (*Citrus maxima*), which is related to lignin deposition in the pulp. Using WGCNA, a module significantly correlated with lignin deposition contained 11 DEGs related to lignin biosynthesis and, more importantly, several TFs showing a high degree of correlation with lignin biosynthesis, among which coding genes for MYB, NAC, OFP6 and bHLH130 TFs were found, providing potential candidate genes to control the onset of this disorder [111]. In another fruit crop, pear, co-expression network analysis contributed to the identification of PpPIF8 as a key regulator of anthocyanin biosynthesis. This gene was rapidly regulated by light and through additional studies, such as overexpression in pear peel and calli and Y2H, confirming its role in anthocyanin biosynthesis and also clarifying its mechanism of action [112]. Nicotine biosynthesis in tobacco was also investigated with WGCNA using different varieties with high and low metabolite content [105]. In this work, co-expressed modules were correlated with nicotine accumulation as an external trait, and genes associated to this module were related to metabolism of nicotine precursors such as Arg, Orn, Asp, Pro and GSH. Hence, elevated levels of these precursors were always related to high nicotine levels. Interestingly, nicotine biosynthesis requires precursors that are also used for polyamine biosynthesis, putrescine being a core intermediate in nicotine biosynthesis and establishing a flow between biosynthetic pathways as a potential way for variety selection. The biosynthesis of tartaric acid in grapes was recently investigated using an in silico approach as the end-product of ascorbate catabolism, for which little information about its metabolism in plants exists. Therefore, taking advantage of public repositories of omics data VTC-Agg (https://sites.google.com/view/vtc-agg, accessed on 20 December 2022), a search for genes involved in this pathway was performed, essentially by interrogating datasets using two already characterized gene candidates (Vv2kgr and VvLidh3) by classical biochemical approaches. Interestingly, these two genes were not mutually co-expressed throughout more than 1300 samples and 33 experiments but were, respectively, co-expressed with other genes involved in ascorbate metabolism, indicating that these two genes are likely not co-regulated and add more complexity to the biosynthetic pathway of tartaric acid, potentially involving plant hormones such as auxin or abscisic acid [113].

## 6. Future Prospects

The advent of single-cell omics, spatial transcriptomics and mass spectrometry imaging (MSI) techniques opens up a new scenario for the integration of omics data in neighboring tissues, contributing to better understand cell-to-cell communication within and between tissues. This will lead to potentially new signaling molecules, including metabolites and proteins, with a role in the integration of exogenous or endogenous signals to develop a particular tissue response. Single-cell RNA sequencing (scRNA-seq) has already contributed to unravel, at least partially, novel gene functions. In Arabidopsis, the investigation of cell-type-specific expression patterns of TMO5/LHW-induced genes in response to phosphate starvation revealed their connection to cytokinin biosynthesis in vascular cells, resulting in an increase in root hair density and phosphate uptake, as well as the identification of cell-type marker genes as responsible for yield traits in maize (reviewed in [52]). To help bridge the spatial gap, different procedures have been implemented to analyze gene expression patterns in specific cell types or histologically defined tissues. Unfortunately, this requires transgenic plants to be generated to exploit cell-specific reporter gene tagging [114,115]. This can be partially overcome by using ultra-thin tissue sections and Laser Capture Microdissection (LCM), but this approach is technically challenging and tedious to obtain sufficient material for RNA extraction. More recently, spatial transcriptomics, which allows the visualization of transcriptome-wide gene expression information in tissue cryosections, achieved using barcoded oligo dT arrays and next-generation sequencing, was developed and its applicability to a wide range of species confirmed [114]. Unfortunately, at present, this methodology does not have resolution at the cell level, but it will surely improve with time and constitutes an interesting strategy to consider when spatial resolution is required. Moreover, the spatial resolution variable can be used in correlation-based approaches to obtain an organ-based interactome in combination with other omics, such as metabolomics. To this respect, MSI strategies [116,117] constitute a suitable option to integrate the spatial variable for plant hormones and metabolites with gene expression.

## Figures and Tables

**Figure 2 ijms-24-02526-f002:**
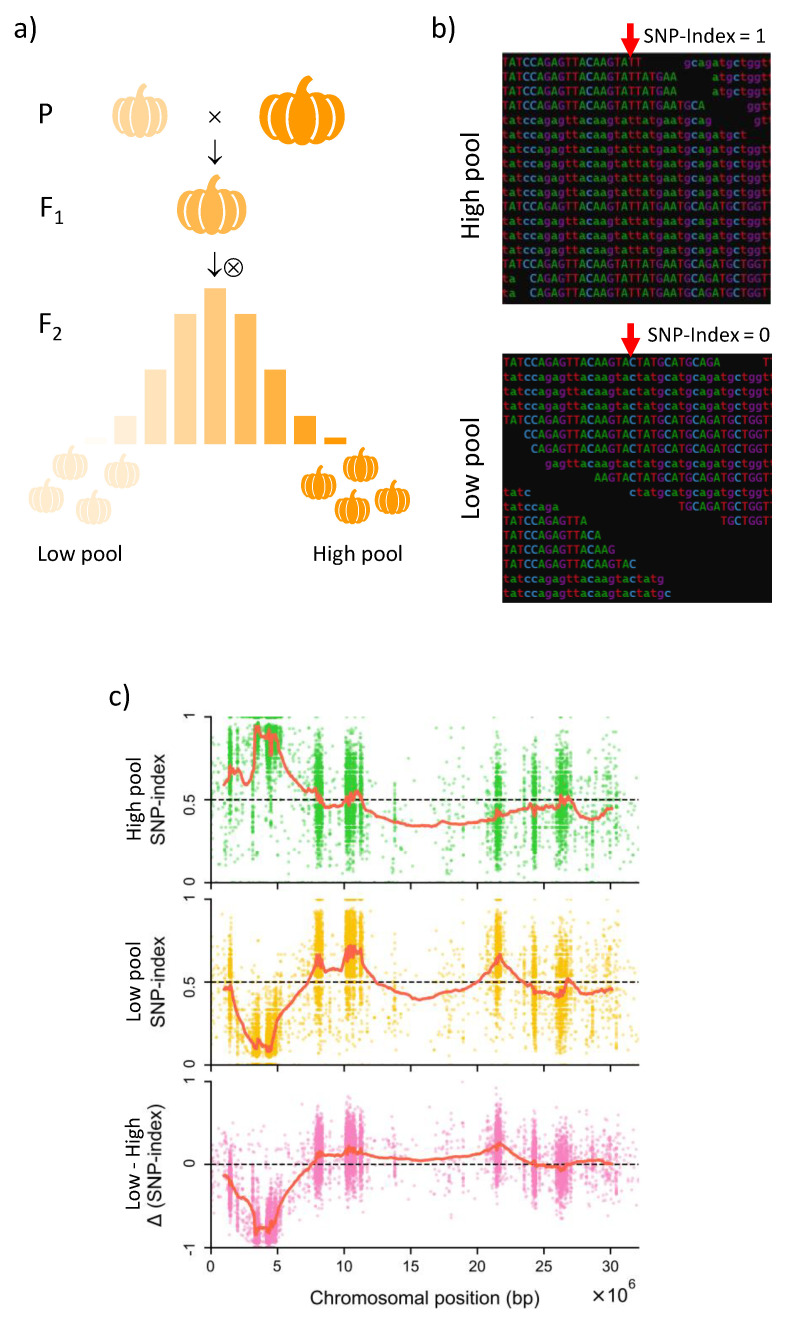
Schematic representation of a QTL-seq experiment to identify SNP loci associated with a particular quantitative trait. (**a**) Plants displaying extreme values (high or low) for a quantitative trait are selected from among the plants of a segregating population. (**b**) Genomic DNA from the plants selected is bulked and sequenced. An alignment of the reads to the reference allows calculating the SNP indices for individual SNPs in the two pools. The red arrows mark the site of one such SNP. (**c**) The SNP indices (allele frequencies) are calculated in the low and high pools for all the available SNP markers along the genome sequence. The presence of a QTL is inferred where the difference between these frequencies, or Δ(SNP-index), significantly deviates from zero.

**Figure 3 ijms-24-02526-f003:**
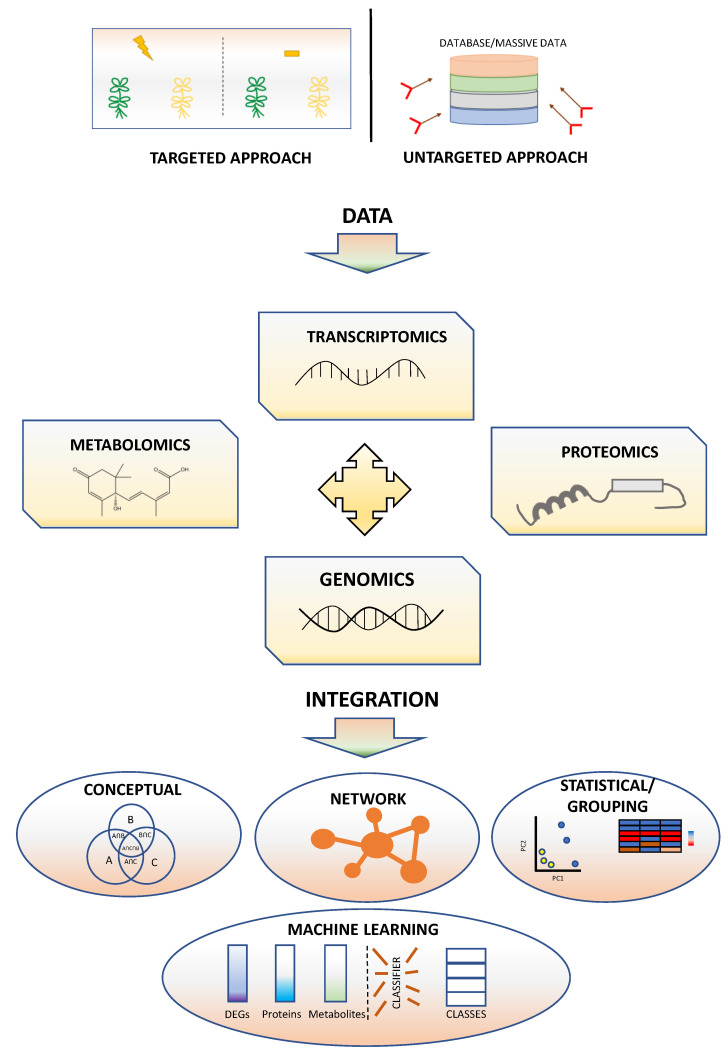
Integration strategies for multiomics data [52].

**Table 1 ijms-24-02526-t001:** Data repositories. All websites accessed on 18 December 2022.

Repository	Functionalities	Species	Web	Reference
SRA	Transcriptomics	All branches of life	https://www.ncbi.nlm.nih.gov/sra	[54]
Metabolights	Raw metabolomics	All branches of life	https://www.ebi.ac.uk/metabolights/	[55]
STRingDB	Protein–protein interactions	All branches of life	https://string-db.org/	[56]
PlaNet	Co-function networks	Photosynthetic organisms	http://aranet.mpimp-golm.mpg.de/	[57]
PGP	Genomics and Phenomics	Chloroplastida ^1^	https://edal-pgp.ipk-gatersleben.de/	[58]
KEGG	Molecular networks	All branches of life	https://www.genome.jp/kegg/	[59]
Reactome	Pathway knowledge	Animalia	https://reactome.org/	[60]
iRefWeb	Protein-protein interactions	All branches of life	http://wodaklab.org/iRefWeb	[61]
GeneMANIA	Gene function	Animalia, Fungi and Plantae	http://genemania.org/	[62]

^1^ Monophyletic group of green plants that includes all land plants (embryophytes) and all green algae (chlorophytes and streptophytes).

**Table 2 ijms-24-02526-t002:** Main software packages used for the integration of omics datasets. All websites were accessed on 20 December 2022.

Software	Omics ^1^	Functionalities	Comments	Repositories	Reference
mixOmics ggmixOmics	T, P, M, R	Multivariant-based framework (PCA, CCA, PLS-DA, etc.)	Dimensions reduction, extraction of variable subgroups connected with traits and visualizations	R/CRAN	[80]
xMWAS	T, P, M	Multivariant- and network-based framework	Application for paired and unpaired study R/GitHub	R/GitHub	[81]
metaboGSE	T, M	Connection of network-based approaches and gene set enrichment analysis	Creation of subnetworks in the context of experimental condition	R/CRAN	[82]
FELLA	M	Network-based enrichment analysis of metabolites lists	Supporting KEGG database	R/BioC	[83]
MetExplore	T, P, M	Network-based analysis, pathway mapping, flux balance modeling and analysis	Easy way for network creation, visualization, curation and metabolite mapping	https://metexplore.toulouse.inrae.fr/index.html/	[84]
OmicsNet	T, P, M, R	Network- and pathway-based approach	Building, visualization and exploration of biological networks in 3D space	https://www.omicsnet.ca/	[85]
MiBiOmics	T, P, M	Correlation-based tool for creating, dimensionality reduction and exploration of networks	Provide the tools for data processing (filtration, normalization and transformation)	https://shiny-bird.univ-nantes.fr/app/Mibiomics	[86]
MetaBridge	T, M	Network-based pathway mapping	Identification of connections between metabolites and enzymes, visualization of data and results	https://metabridge.org/	[87]

^1^ T, transcriptomics; P, proteomics; M, metabolomics; R, regulatory omics.

## Data Availability

Not applicable.

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
