# Peer review of "From Classical to Modern Computational Approaches to Identify Key Genetic Regulatory Components in Plant Biology"

_ijms, 2023, doi:10.3390/ijms24032526_

Round 1
Reviewer 1 Report
The authors reviewed the modern technology of genomics, transcriptomics, metabolomics, proteomics, and those network analysis which help to reveal a genetic basis of plant biology. This manuscript has potentially provided an important contribution to the rapidly growing field of omics analysis.
However, I have some concerns about the overall structure of the manuscript. Although this manuscript presents a large amount of information on related tools of omics analysis, it is poorly summarized and difficult to read. In addition, an there are many errors, the author should carefully check the text. Therefore, I cannot recommend the publication of this manuscript at this stage. Listed below are my specific comments. These comments will be helpful.
Comments
1. Overall, this manuscript is poorly summarized the recent technologies and their related tools, and it is not easy to read. Please make tables of tools/software for each omics analysis the authors describe in the main text and reduce the text volume.
2. Please make sure you have permission from the publisher/author to use Figures 1 and 2.
3. Page 1, L26, “…yield-, stress or quality-related plant traits..”. The word "plant" can be deleted.
4. Page 1, L35, “…to specific phenotype traits (canopy….”. It is better to change to "phenotype" or "phenotypic traits".
5. Page 2, L49, "2.1. Where it all began: Quantitative trait loci or QTLs". It is better to change "Quantitative trait loci (QTLs)".
6. Page 2, L59, “…either allowed to self-pollinate”. It is better to change to "self-pollinate with single seed descent method to develop recombinant inbred lines (RILs)".
7. Page 2, L65, “…e.g. the IL collection of …”. Please add the description of IL.
8. Page 2, L75, “…phenotypic value[4]). Markers…”. Delete ")".
9. Page 2, L78-80, This sentence is an oversimplification. Important traits such as yield, for example, are made up of a large number of QTLs.
10. Page 2, L88-9, Please add appropriate references.
11. Page 3, L121, “…trait of interest [17]), help…”. Delete ")".
12. Page 4, L127, “…markers is tested. ). More…”. Delete ")".
13. Page 4, L140, “…QTL-seq, or genome-wide association…”. The word "GWAS" already exists.
14. Page 4, L144, QTL-seq also have to measure the phenotype of a large number of individuals.
15. Page 4, L146, The word "NGS" already exists.
16. Page 4, L161, It is better to change to "RNA-seq".
17. Page 4, L198, "…individual QTLs into near-isogenic lines (NILs)." It is better to change "QTLs into an elite parent line to develop the near-isogenic lines (NILs)."
18. Page 6, L222, The word "NGS" already exists.
19. Page 6, L228, "…from the 60 megabases (Mb) size of...". It is better to change to "60 Mb".
20. Page 6, L245-6, "…~32-fold increase in the mean contig N50 (the length...". How about adding the phrase "than using short-read technologies"?
21. The authors should refer to table 1 in the main text.
22. Page 11, L391-409, These sentences may be deleted or at least should be shortened.
23. Please check the inserted position of Figure 3 is correct.
Author Response
The authors reviewed the modern technology of genomics, transcriptomics, metabolomics, proteomics, and those network analysis which help to reveal a genetic basis of plant biology. This manuscript has potentially provided an important contribution to the rapidly growing field of omics analysis.
However, I have some concerns about the overall structure of the manuscript. Although this manuscript presents a large amount of information on related tools of omics analysis, it is poorly summarized and difficult to read. In addition, an there are many errors, the author should carefully check the text. Therefore, I cannot recommend the publication of this manuscript at this stage. Listed below are my specific comments. These comments will be helpful.
Many thanks for taking the time to review our work! We have tried to implemented all changes suggested in order to improve readability and corrected all error highlighted.
Comments
- Overall, this manuscript is poorly summarized the recent technologies and their related tools, and it is not easy to read. Please make tables of tools/software for each omics analysis the authors describe in the main text and reduce the text volume.
Thanks for your suggestion. Actually, the manuscript focuses on the strategies/tools to analyze omics data not the generation of omics data per se. Nevertheless, we understand your criticism is related to the outraging text volume, therefore we have shortened this and other sections following your indications. Hope now the manuscript is more clear and easier to follow.
- Please make sure you have permission from the publisher/author to use Figures 1 and 2.
The figures were completely rebuilt using the cited references as inspiration, they are completely new to avoid any comparison with previously published and copyrighted material.
- Page 1, L26, “…yield-, stress or quality-related plant traits..”. The word "plant" can be deleted.
Thanks! We have deleted the word ‘plant’.
- Page 1, L35, “…to specific phenotype traits (canopy….”. It is better to change to "phenotype" or "phenotypic traits".
Thanks! We have replaced the word ‘phenotype’ by ‘phenotypic’.
5. Page 2, L49, "2.1. Where it all began: Quantitative trait loci or QTLs". It is better to change "Quantitative trait loci (QTLs)".
Thanks again for the suggestion. We have amended the section title.
- Page 2, L59, “…either allowed to self-pollinate”. It is better to change to "self-pollinate with single seed descent method to develop recombinant inbred lines (RILs)".
Many thanks for the suggestion, we have replaced the entire sentence.
- Page 2, L65, “…e.g. the IL collection of …”. Please add the description of IL.
Thanks, we have made the amendment.
- Page 2, L75, “…phenotypic value[4]). Markers…”. Delete ")".
We apologize for this mistake, that parenthesis skipped the successive rounds of revision. Thanks for your minute revision.
- Page 2, L78-80, This sentence is an oversimplification. Important traits such as yield, for example, are made up of a large number of QTLs.
We agree with your perception and we have added the following section to the sentence, to better clarify this point:
“...although the latter is the most usual situation, making marker development a daunting task”
- Page 2, L88-9, Please add appropriate references.
We have added a reference to support the statement in the sentence below, and we have modified the text to be more specific.“Typically, this mutagenic compound generates random point mutations that differ from one crop to another, 1 mutation per Mb in barley, to 1 mutation per 175 kb in Arabidopsis or per 25 kb in hexaploid wheat [12] and has been widely used in forward genetics as a source of random variability arising from a highly homogeneous population (e.g. seeds from a single Arabidopsis plant, a cell culture obtained from a single genotype and tissue, etc.) [13].”
- Arisha, M.H.; Shah, S.N.M.; Gong, Z.H.; Jing, H.; Li, C.; Zhang, H.X. Ethyl Methane Sulfonate Induced Mutations in M2 Generation and Physiological Variations in M1 Generation of Peppers (Capsicum Annuum L.). Front. Plant Sci. 2015, 6, 1–11, doi:10.3389/fpls.2015.00399
- Candela, H.; Hake, S. The art and design of genetic screens: Maize. Nat. Rev. Genet. 2008, 9, 192–203, doi:10.1038/nrg2291.
11. Page 3, L121, “…trait of interest [17]), help…”. Delete ")".
We apologize for this mistake, that parenthesis skipped the successive rounds of revision. Thanks for your minute revision.
- Page 4, L127, “…markers is tested. ). More…”. Delete ")".
We apologize for this mistake, that parenthesis skipped the successive rounds of revision. Thanks for your minute revision.
- Page 4, L140, “…QTL-seq, or genome-wide association…”. The word "GWAS" already exists.
We apologize for this mistake, we have removed the full denomination of the approach and left GWAS only.
- Page 4, L144, QTL-seq also have to measure the phenotype of a large number of individuals.
We have added this part to the sentence. Thanks for the suggestion.
- Page 4, L146, The word "NGS" already exists.
We have replaced the full denomination of the approach by NGS.
- Page 4, L161, It is better to change to "RNA-seq".
We have replaced RNAseq by RNA-seq where necessary throughout the text.
- Page 4, L198, "…individual QTLs into near-isogenic lines (NILs)." It is better to change "QTLs into an elite parent line to develop the near-isogenic lines (NILs)."
Thanks! We have made the amendment.
- Page 6, L222, The word "NGS" already exists.
Many thanks! We have replaced new generation sequencing by NGS.
- Page 6, L228, "…from the 60 megabases (Mb) size of...". It is better to change to "60 Mb".
Thanks! We have made the amendment.
- Page 6, L245-6, "…~32-fold increase in the mean contig N50 (the length...". How about adding the phrase "than using short-read technologies"?
Thanks! We have made the amendment.
- The authors should refer to table 1 in the main text.
We are sorry for the mistake, we have mentioned table 1 in P9 L398 and P10 L414.
- Page 11, L391-409, These sentences may be deleted or at least should be shortened.
After careful inspection, we have shortened or deleted sentences in the section indicated.
- Please check the inserted position of Figure 3 is correct.
We have moved Figure 3 to P12 which makes more sense. Thanks for the suggestion.
Reviewer 2 Report
The authors present a comprehensive review of the relevant methods to identify key components in plant biology. Beginning with the now classical QTL estimation, they take the reader through a very well explained tour visiting the fields of marker assisted selection, comparative genomics and correlation of genes and traits, finishing with a remarkably enlightening explanation of network estimation.
I highly appreciated the easy to follow explanation about the different methods and software available; a field that is difficult to navigate, given the number and complexity of the different approaches.
In section "2.2. Marked assisted selection", I suggest to add a comment in the sense that strategies using molecular markers (in any of their forms) will benefit by the parallel use of classical phenotypic selection, possibly through the use of weighted selection indexes. This is due to the fact that no molecular method will explain 100% of the total variance of a target trait and also there could be pleiotropic effects that are difficult to take into account. However, this suggestion is not compulsory.
I could not find the place where "Table 1" (data repositories) was mentioned in the text. It is necessary to add a sentence referring to that table (which is highly relevant).
In some instances the word "regard" could be used instead of "respect" (but this suggestion needs to be evaluated case by case).
Minor suggestions:
(line number followed by suggestion for edition)
75: Extra ")" after "[4]", no space between "value" and "[4]".
121: same problem: "[17])," must be "[17],".
127: ". )." must be ".".
151: "[25] ." correct to "[25]."
160: "[26–28])." must be "[26–28]."
178: Consider "regulatory proteins. Such transcription factors" instead of "regulatory proteins, such transcription factors".
192: a space is needed after the capital delta symbol.
449: Substitute "[68] " by "[68]. "
452: change ", there is no definitive and standardized methodology, up to date." to ", up to date there is no definitive and standardized methodology"
536: Considering changing "and inference methods." to "and inference methods exist." or "and inference methods are available."
684: Close the parenthesis after "[102]."; i.e., "[102])."
Author Response
I highly appreciated the easy to follow explanation about the different methods and software available; a field that is difficult to navigate, given the number and complexity of the different approaches.
In section "2.2. Marked assisted selection", I suggest to add a comment in the sense that strategies using molecular markers (in any of their forms) will benefit by the parallel use of classical phenotypic selection, possibly through the use of weighted selection indexes. This is due to the fact that no molecular method will explain 100% of the total variance of a target trait and also there could be pleiotropic effects that are difficult to take into account. However, this suggestion is not compulsory.
Many thanks for taking the time to review our work! We have tried, to the best of our knowledge, to address all concerns raised and to implement all changes suggested to improve our manuscript.
In regard to the reviewer’s observation, we agree and we have indicated this in the text L232-236.
“Moreover, using molecular markers (in any of their forms) will benefit from the parallel use of classical phenotypic selection, possibly through the use of weighted selection indexes, because no molecular marker will explain 100% of the total variance of a target trait and there might be pleiotropic effects that are difficult to take into account.”
I could not find the place where "Table 1" (data repositories) was mentioned in the text. It is necessary to add a sentence referring to that table (which is highly relevant).
We are sorry for the mistake, we have mentioned table 1 in P9 L398 and P10 L414.
In some instances the word "regard" could be used instead of "respect" (but this suggestion needs to be evaluated case by case).
Thanks. After careful inspection, we have replaced the word respect by the word regard in L450, L534 and L695.
Minor suggestions:
(line number followed by suggestion for edition)
75: Extra ")" after "[4]", no space between "value" and "[4]".
We apologize for this mistake that skipped the successive rounds of revision. Thanks for your minute revision
121: same problem: "[17])," must be "[17],".
We apologize for this mistake that skipped the successive rounds of revision. Thanks for your precise revision
127: ". )." must be ".".
We apologize for this mistake, that parenthesis skipped the successive rounds of revision. Thanks for your precise revision.
151: "[25] ." correct to "[25]."
We apologize for this mistake that skipped the successive rounds of revision. Thanks for your precise revision
160: "[26–28])." must be "[26–28]."
We apologize for this mistake, that parenthesis skipped the successive rounds of revision. Thanks for your precise revision.
178: Consider "regulatory proteins. Such transcription factors" instead of "regulatory proteins, such transcription factors".
Thanks. After careful inspection, we have rewritten the sentence.
192: a space is needed after the capital delta symbol.
449: Substitute "[68] " by "[68]. "
Thanks! We have made the amendment.
452: change ", there is no definitive and standardized methodology, up to date." to ", up to date there is no definitive and standardized methodology"
Thanks! We have made the amendment.
536: Considering changing "and inference methods." to "and inference methods exist." or "and inference methods are available."
Thanks! After careful inspection, we have made the amendment.
684: Close the parenthesis after "[102]."; i.e., "[102])."
Thanks! We have made the amendment.
Round 2
Reviewer 1 Report
The authors have adequately addressed all the points raised in the first version of the manuscript, and I am satisfied with the authors' revisions.
Thus, I recommend this manuscript for publication in IJMS.